# Evaluating the Performance of the WHO International Reference Standard for Osteoporosis Diagnosis in Postmenopausal Women of Varied Polygenic Score and Race

**DOI:** 10.3390/jcm9020499

**Published:** 2020-02-12

**Authors:** Qing Wu, Xiangxue Xiao, Yingke Xu

**Affiliations:** 1Nevada Institute of Personalized Medicine, University of Nevada, Las Vegas, NV 89154, USA; xiangxue.xiao@unlv.edu (X.X.); yingke.xu@unlv.edu (Y.X.); 2Department of Environmental and Occupational Health, School of Public Health, University of Nevada Las Vegas, NV 89154, USA

**Keywords:** polygenic score (PGS), bone mineral density (BMD), single nucleotide polymorphism (SNP)

## Abstract

Background: Whether the bone mineral density (BMD) T-score performs differently in osteoporosis classification in women of different genetic profiling and race background remains unclear. Methods: The genomic data in the Women’s Health Initiative study was analyzed (n = 2417). The polygenic score (PGS) was calculated from 63 BMD-associated single nucleotide polymorphisms (SNPs) for each participant. The World Health Organization′s (WHO) definition of osteoporosis (BMD T-score ≤ −2.5) was used to estimate the cumulative incidence of fracture. Results: T-score classification significantly underestimated the risk of major osteoporotic fracture (MOF) in the WHI study. An enormous underestimation was observed in African American women (POR: 0.52, 95% CI: 0.30–0.83) and in women with low PGS (predicted/observed ratio [POR]: 0.43, 95% CI: 0.28–0.64). Compared to Caucasian women, African American, African Indian, and Hispanic women respectively had a 59%, 41%, and 55% lower hazard of MOF after the T-score was adjusted for. The results were similar when used for any fractures. Conclusions: Our study suggested the BMD T-score performance varies significantly by race in postmenopausal women.

## 1. Introduction

Osteoporosis is a common, progressive systemic skeletal disease characterized by low bone mass and deteriorated bone tissue, resulting in an increase in bone fragility and susceptibility to fracture [1]. Osteoporosis-associated fractures often cause a significant increase in morbidity, mortality, and accompanying social and economic costs [2]. By estimation, about 50% of postmenopausal Caucasian women and 20% of Caucasian men in the US will suffer at least one fragility fracture after the age of 50 [3]. With life expectancy increasing universally, osteoporosis and fracture will become an ever-growing health problem worldwide [4].

Osteoporosis is a silent disease because bone loss occurs or bone tissue deteriorates without any symptoms [4]. Patients often are not aware that they have osteoporosis until a fracture occurs. Thus, correctly diagnosing osteoporosis and identifying individuals who will sustain osteoporotic fracture is critical for the prevention of devastating fracture outcomes in the aging population. As BMD is the single strongest predictor of primary osteoporotic fracture [5], clinical osteoporosis diagnosis is based on BMD measurements from dual-energy x-ray absorptiometry (DEXA) assessment [6]. The World Health Organization established commonly accepted definitions of osteoporosis as a femur neck BMD that lies ≥2.5 standard deviations below (T-score ≤ −2.5) the mean value for young, healthy women [7]. This definition becomes the WHO international reference standard for osteoporosis diagnosis. However, the major limitation of this WHO reference standard is its low sensitivity; most fractures occur in individuals with a femur neck BMD T-score > −2.5. In addition, because many other risk factors, including age, female gender, and previous fracture, are associated with fracture risk independently of BMD, several predictive models have been developed to estimate fracture risk from these established risk factors. The Fracture Risk Assessment Tool (FRAX) is the most commonly used fracture risk assessment tool in the US [8]. Although FRAX improves fracture prediction over the BMD T-score method alone, the predictive performance of both FRAX and WHO T-score method varies in different population cohorts [8,9] and with different conditions [10,11].

The original FRAX was developed from nine large cohorts and then validated in 11 independent cohorts across the world [1]. The US FRAX was calibrated from the data of the Rochester Epidemiology Project [12], composed predominantly of Caucasians [13]. Further, the T-score was initially proposed only for postmenopausal Caucasian women [14,15]. Although both FRAX and T-score were adjusted subsequently for race and ethnicity, the methodology for the adjustment was not empirically based, thus rendering their performance for fracture prediction unreliable in minorities. In addition, neither FRAX nor T-score takes account of genetic components even though research has shown that hereditary factors are determinants of bone structure and are strongly associate with bone mass decrease, bone deterioration, and fragility fractures. With the development of advanced genomic technologies, numerous genetic loci related to fracture and BMD have been discovered in major genome-wide association studies (GWASs) and genome-wide meta-analyses. Both of these factors provide a unique opportunity to examine the performance of existing clinical fracture prediction approaches in groups with different genomic profiling.

Our previous study has examined the performance of FRAX in postmenopausal women by race and polygenic score, computed from fracture-associated SNPs discovered in the largest GWAS meta-analysis (under review). The T-score method (T-score ≤ −2.5) is the WHO international reference standard for osteoporosis diagnosis and has been endorsed by numerous professional societies, including the International Society of Clinical Densitometry (ISCD) [16], and widely used in clinical practice for osteoporosis diagnosis. However, the performance of the T-score method for osteoporosis classification in the U.S. minorities was rarely studied, and the T-score performance in osteoporosis classification with different genetic profiling has never been reported in the literature. Thus, this study aimed first to evaluate whether T-score performs differently in osteoporosis classification with different polygenic risk scores, and second to assess T-score performance in osteoporosis diagnosis by race in women. We also examined the extent to which the interaction of race and polygene scores impacts the T-score performance in osteoporosis classification and fracture prediction.

## 2. Experimental Section

### 2.1. Data Source

The Women’s Health Initiative (WHI) study is a nationwide longitudinal study to examine the health of postmenopausal women aged 50–79 years old who have no severe medical conditions at baseline [17]. Between 1993 and 1998, the WHI enrolled 161,809 women aged 50 to 79 years old at 40 clinical centers nationwide. The details of WHI recruitment and follow-up procedures have been described elsewhere [17]. Briefly, eligible women were enrolled in one or more randomized Clinical Trials (CT) or to an Observational Study (OS). Participants were followed up on by mail or telephone semiannually in CT and with questionnaires annually in the OS. The Institutional Review Boards at each participating institution approved the study protocols and participant consent forms [17].

### 2.2. Participants

The data used for the present study were de-identified and were acquired through the database of Genotype and Phenotype (dbGap) (https://www.ncbi.nlm.nih.gov/projects/gap/cgi-bin/study.cgi?study_id=phs000200.v12.p3) with the approval of the institutional review board at the University of Nevada, Las Vegas. The data included in this analysis were merged from four WHI sub-studies, WHI Genomics and Randomized Trials Network (GARNET), National Heart Lung, and Blood Institute (NHLBI), Population Architecture using Genomics and Epidemiology (PAGE), and Women’s Health Initiative Memory Study (WHIMS). On baseline questionnaires, participants provided information on age, race/ethnicity, smoking, physical activity, and supplement use. The included subjects were genotyped using either the Affymetrix 6.0 array set (Affymetrix, Santa Clara, CA, USA) or the Illumina (Illumina, San Diego, CA, USA) platform. Participants who reported taking any medication known to influence osteoporosis, including corticosteroid bisphosphonates, calcitonin, parathyroid hormone, selective estrogen receptor modulators, luteinizing hormone-releasing hormone agents, and somatostatin agents, as well as participants who did not have BMD measurements were excluded from the analytic sample. In total, there were 2417 eligible participants from multiple ethnic backgrounds, with genotype data and adjudicated fracture outcomes available.

### 2.3. BMD Measurements

BMD was measured for women using dual-energy X-ray absorptiometry (QDR 2000 or 2000+, or 4500 W; Hologic Inc, Bedford, MA, USA) at three participating US clinical centers (Birmingham, AL; Pittsburgh, PA; and Tucson/Phoenix, AZ). Certified technicians using standard protocols measured BMD of the total hip, lumbar spine (L2–L4), and total body. Baseline BMD measurements were employed to classify participants in order to determine if they have osteoporosis at baseline in this study. Standard quality assurance protocols for positioning and analysis, routine hip and spine phantoms, and review of a randomly selected sample were employed. Changes of hardware and software were centralized and calibrated, and calibration phantoms across instruments and clinical sites were in close agreement, with inter-scanner variability <1.5% for the spine, <4.8% for the hip, and <1.7% for linearity [18]. BMD T-scores were calculated for each individual by using the young adult, normal Caucasian women reference databases. Osteoporosis was defined as a BMD that lies 2.5 standard deviations or less below the average value for young, healthy women [19].

### 2.4. Outcomes: Incident Fractures

In this study, any fractures were defined as all fractures except those of fingers, toes, ribs, sternum (or chest), skull (or face), and cervical vertebrae. Major osteoporotic fracture (MOF) was defined as a composite of hip, humerus, forearm, and clinical vertebral fractures. The study participants were followed for 19 years, from the inception of WHI initial study to the end of the WHI extension study II, to ensure a sufficiently long follow-up duration to capture enough events. The follow-up period was computed from the date of the enrollment (OS) or randomization (CT) to the time of the first fracture or the time of death. Self-reported fracture outcomes were identified annually in OS and semiannually in CT by questionnaires. All fractures in the CT and hip fractures in the OS were adjudicated by using radiology reports. Hip fractures were adjudicated centrally or locally using the same criteria. The agreement between central and local adjudication was 96% for hip fractures [20]. Other types of fractures were adjudicated locally at the clinical centers which were not designed for BMD measurements in the WHI study [17].

### 2.5. Genotyping

Blood samples were genotyped using genomic DNA for WHI participants. Genomic data of WHI were acquired through dbGap. Genotype imputation was conducted at the Sanger Imputation Server to impute variants that were missing, un-typed, or poorly captured in the original data. The Haplotype Reference Consortium (HRC) reference panel and Positional Burrows-Wheeler Transform (PBWT) imputation algorithm were employed for genotype imputation. All 63 fracture-associated SNPs reported by Estrada et al. [21] were successfully imputed. The imputation quality was high, with R^2^ = 0.99.

### 2.6. Polygenic Score

Genetic risk for decreased BMD was quantified using a standardized metric described in detail by Estrada et al. [21]. Briefly, this metric allows the composite assessment of genetic risk for complex traits by summarizing the genetic predisposition. Based on 63 femoral neck BMD-associated SNPs discovered in the largest genome-wide meta-analysis [21], the polygenic score was computed as PGS = sum (x_i_ × b_i_); where x_i_ are individual’s genotype (0, 1, 2) for SNP i, and b_i_ are the effect size of this SNP. Linkage disequilibrium (LD) pruning was executed in advance to remove possible LD that existed between SNPs. None of the 63 SNPs were deleted after pruning. To demonstrate if the performance of the WHO international reference standard for osteoporosis diagnosis varied by PGS, eligible WHI participants were divided into three PGS groups using distribution of 25%, 50%, and 25%.

### 2.7. Statistical Analysis

Demographic and baseline clinical characteristics are presented as mean ± SD for continuous variables or frequencies (%) for categorical variables. Differences between the individuals with and without a fracture were examined by using Student’s t-test for continuous variables and by using chi-square tests or Fisher’s exact test (when numbers were small) for categorical variables, respectively. PGS in different races was examined by using ANOVA. The observed cumulative incidence of fracture from the start of WHI to the end of WHI extended II was assessed by race and PGS groups. The cumulative incidence function (CIF) was applied to derive the observed fracture probability for MOF and any fracture with competing mortality risk accounted for. The ratio between T-score predicted fracture incidence and observed fracture incidence (POR), with the corresponding 95% CI, was calculated for each subgroup.

To assess the performance of the T-score method in classifying osteoporosis in different subgroups, the false-positive rate, and the false-negative rate was calculated for each PGS and race group. Multivariate Cox Proportional Hazard Model was employed to assess the effect of PGS and race on the outcome of MOF or any fracture within 19 years, with baseline T-score controlled for. To further assess whether the effect of PGS and race on the outcome of MOF or any fracture are independent of other common risk factors of osteoporosis, separate multivariate Cox Proportional Hazard Models were conducted with baseline T-score, age, body mass index (BMI), and previous fracture controlled for. The T-score diagnosis was treated as a binary categorical variable in the Cox Proportional Hazard Model. Considering that the PGS used in the present study was calculated based on femoral neck BMD-related SNPs, we also assessed whether the predictive value of PGS for hip fracture would be different from other types of fracture. A multinomial logistic regression with three outcomes (hip fracture, non-hip fracture, non-fracture) was performed. 

A series of sensitivity analyses were also conducted, with the first one was conducted on a small sample (N = 1775) in which participants who had previous fractures were excluded. To be comparable with FRAX, which assess the 10-year probability of MOF, the POR between predicted fracture incidence and observed incidence of MOF and any fracture in 10 years were also assessed, along with the false-positive rate and false-negative rate for MOF and any fracture classification in different PGS and race groups with 10-year follow up. A subgroup analysis was conducted to assess whether PGS would predict fracture differently in osteopenia patients, and participants with normal BMD at baseline. Statistical analyses were performed using the SAS 9.4 (SAS Institute, Inc., Cary, NC, USA).

## 3. Results

### 3.1. Baseline Characteristics

The study included a total of 2417 women for analysis. During the 19-year follow-up period, 634 (26.23%) women died, and 289 (11.96%) women sustained at least one fracture at any skeletal site during the follow-up. There were 52 women free of the previous fracture had a T-score diagnosis of osteoporosis at baseline, and 12 of them had a new fracture during the follow-up. Table 1 compares the baseline characteristics of women with and without any fracture during the follow-up. Women who sustained a fracture were older (*p* < 0.001), had lower body mass index (BMI), higher prevalence of prior fractures (*p* < 0.001), and more hip fractures in their family history (*p* = 0.002). T-score was significantly lower in women with a fracture incidence (*p* < 0.0001). PGS was not significantly different between women who sustained a fracture and women who did not (*p* = 0.81), yet was significantly different between race groups (*p* < 0.0001).

### 3.2. Performance of T-Score in Predicting MOF and Any Fracture

The T-score prediction versus the observed cumulative incidence of MOF and any fracture during the 19-year follow-up period by PGS groups are shown in Figure 1A. The 19-year MOF incidence derived from T-score significantly underestimated risk across all PGS groups. The most significant underestimation by T-score was observed in women who had low PGS, in which the cumulative incidence of MOF was 3.83% versus observed 8.8%, with a corresponding predicted/observed ratio (POR) of 0.43 (95% CI, 0.28–0.64), followed by the medium PGS group with a POR of 0.71 (95% CI, 0.56–0.90); and in the high PGS group, the POR was 0.72 (95% CI, 0.52–0.98). Similar results were also observed when using any fracture as the outcome, the estimated incidence calculated by T-score underestimated fracture risk in all PGS groups (Figure 1B).

The predicted versus the observed cumulative incidence of MOF and any fracture by racial groups are shown in Figure 2. The T-score estimated incidence of MOF underestimated fracture risk in all racial groups except American Indians, with the statistically significant underestimation only observed in African American and Caucasian women. In African American women, the predicted incidence of MOF was 1.42% versus observed 2.73%, and the POR was 0.52 (95% CI 0.30–0.83). In Caucasian women, the predicted incidence of MOF was 11.55%, as opposed to observed 18.38%, with the POR being 0.63 (95% CI 0.50–0.78) (Figure 2A). Similarly in any fracture, significant underestimation of the fracture incidence was observed in African American (POR: 0.19, 95% CI: 0.11–0.31), Hispanic (POR: 0.48, 95% CI: 0.33–0.67), and Caucasian women (POR: 0.47, 95% CI: 0.37–0.59) (Figure 2B).

Figure 3 demonstrates the false positive rate and false-negative rate for MOF and any fracture based on the T-score classification by the PGS group. By defining a T-score < −2.5 as the presence of osteoporosis, for MOF prediction, the false positive rate of this test was low across all PGS groups, with the lowest sensitivity of T-score observed among women who have low genetic risk of low bone mass (3.58%). The false-negative rate was highest in the low PGS group (92.86%), followed by the high PGS group (85.42%), and medium PGS group (80.95%). Similar results were also observed in the outcome of any fracture.

Figure 4 demonstrates the race-specific false positive rate and false-negative rate for MOF and any fracture based on the T-score classification. By defining a T-score < −2.5 as the presence of osteoporosis, for MOF prediction, the false positive rate of this test was low across all race groups, with the lowest observed among African American women (1.35%), followed by Hispanic women (5.07%), American Indian women (6.6%), and Caucasians (9.8%). The false-negative rate was high across all race groups, with the lowest observed in Caucasians (79.44%) and the highest in African Americans (96.15%). Similar results were also observed for the outcome of any fractures.

### 3.3. PGS and the Fracture Outcome

In the Cox Proportional Hazard Model, after adjusting for baseline T-score classification, weighted PGS calculated from 63 femoral neck BMD-related SNPs was not significantly associated with subsequent MOF. Compared to the low PGS group, the probability of sustaining a MOF was 14% lower for women with medium genetic risk (HR = 0.86, 95% CI 0.68–1.09) and 2% lower for women with high genetic risk (HR = 0.98, 95% CI 0.75–1.28). Similar findings with the outcome of any fracture were observed (Table 2). Results from the multinomial logistic regression indicated that the effect of PGS for predicting hip fracture is not different from the effect of PGS for predicting non-hip fractures (*p* = 0.51) (Table A1). Moreover, the predictive value of PGS in osteopenia participants and women with normal BMD at baseline remained minimal (results not shown).

### 3.4. Race/ethnicity and the Fracture Outcome

After controlling for baseline T-score, race remained a significant predictor of subsequent MOF and any fracture. Compared to Caucasian women, African American women had a 59% lower hazard of MOF (HR = 0.41, 95% CI 0.33–0.52) and 47% lower hazard of any fracture (HR = 0.43, 95% CI 0.32–0.88); American Indian women had a 41% lower hazard of MOF (HR = 0.59, 95% CI 0.35–0.99) and 56% lower hazard of any fracture (HR = 0.44, 95% CI 0.36–0.54); Hispanic women had a 55% lower risk of MOF (HR = 0.55, 95% CI 0.35–0.58) and 54% lower risk of any fracture (HR = 0.46, 95% CI 0.36–0.58). The potential impact of PGS on the estimated risk of MOF and any fractures across different racial groups was also assessed. When adjusted for T-score and PGS group, the impact of race on the estimated probabilities MOF and any fracture was slightly attenuated but remained statistically significant. Similar results were observed when using any fractures as the outcome (Table 3). After adjusting for other common risk factors of osteoporosis, only African American women remained to have a significantly lower hazard of MOF (HR = 0.67, 95% CI 0.52–0.89) and any fracture (HR = 0.71, 95% CI 0.56–0.91), compared with Caucasian women (Table A2).

### 3.5. Sensitivity Analysis

We conducted a sensitivity analysis in which subjects who had previous fractures at baseline were excluded (Table A3 and Table A4). Results of the Cox Proportional Hazard Model remained the same except that after adjustment for T-score diagnosis and PGS, the impact of race on the estimated risk of MOF and any fractures attenuated slightly. Compare to Caucasian women, the adjusted hazard of MOF was 43%, and 55% lower in African-American and Hispanic women, respectively. The hazard of MOF was no longer significant between Caucasian and American Indian women. Similar results were also observed when using any fracture as the outcome. The T-score classification perfomed slightly different when comparing with observed 10-year cumulative incidence of fracture, with the 10-year MOF incidence overestimated the risk of fracture in medium and high PGS groups (Figure A1, Figure A2, Figure A3 and Figure A4).

## 4. Discussion

The present study provides compelling evidence that during the 19-year follow-up, the T-score method underestimates the risk of MOF and any fracture in women 50–79 years old, across all racial and PGS groups, especially in African Americans and women who have a low genetic risk. Moreover, the prognostic performance of the T-score method estimated by false-positive rate and false-negative rate using the cut-off value of −2.5 differed across race and PGS groups as well. Results from the multivariate Cox proportional Models provided further evidence that the performance of the T-score method in predicting osteoporotic fracture risks varies by race.

The BMD threshold defined by the WHO T-score method was found to be problematic. The National Osteoporosis Risk Assessment Study found that 82% of 2259 women who reported fractures had a T-score > −2.5 [22]. Similarly, in the Rotterdam Study of 7806 people, both 56% of women and 79% of men with non-vertebral fractures had a T-score of > −2.5 [23]. Other studies also demonstrated that the majority of low-trauma fractures occur in individuals whose T-scores were above −2.5 [24,25], which is consistent with the extremely high false-negative rates observed in the present study, especially in African American and Hispanic women, as well as women who have a low genetic risk. However, the percentage of being misclassified into a high-risk group without sustaining a fracture is highest among Caucasian women when a T-score method is used to assess fracture risk. BMD is known to be the single best predictor of fracture and the differences have been identified in the areal BMD between ethnic and racial groups [26]. However, the observed cumulative incidence of fracture, in terms of both MOF and any fracture, was significantly higher than the estimation derived from the BMD-based T-score method in minorities. The results of multivariate Cox Proportional Hazard Analysis further demonstrated that race is a significant predictor of MOF and any fracture independent of the T-score classification. Although separate reference database was proposed for Africa Americans and Hispanics [27], we did not use this ethnic-specific references in this study because whether the T-score derived from the ethnic-specific database performs better or worse in osteoporosis diagnosis remains unclear [28]. Nonetheless, our previous study suggested that a new classification method of low BMD based on the race-specific lower limit of normal values may help mitigate some of the T-score limitations in minority populations [29].

The present study found that T-score greatly underestimated the risk of fracture in women aged 50–79 years old, and the degree of underestimation by the T-score method in the low PGS group is greater than in the high genetic risk groups in both outcomes of MOF and any fracture. However, in the multivariate analysis, genetic profiling was demonstrated not to be a significant predictor of MOF and any fracture, after T-score classification was adjusted for. Prior twin studies demonstrated that the heritable component of fracture is largely independent of BMD [30,31], whereas the reported fracture-related genetic variants are also associated with BMD [32]. Due to the study power issue, GWAS for dichotomous disease as a direct outcome has yielded relatively lower numbers of loci discovered, and this is also the concern for osteoporotic fracture studies as well. Moreover, the multifactorial nature of fracture is another issue that makes it challenging to identify the specific genetic determinants that contribute to the risk of fractures. Therefore, the PGS constructed in the present study may not sufficiently capture the BMD-independent genetic risk of fracture. With more fracture-related genetic components being discovered, a more significant effect of PGS on fracture risk prediction should be foreseen. Another possible reason for the minor effect of PGS on fracture outcomes observed in the present study is that, similar to other age-related traits, the heritability of fracture risk decreases with age [32]. Since the analytic sample consisted of older women, the effect mediated through genetic influences on bone turnover, and bone geometry or non-skeletal factors such as cognitive function, neuromuscular control, visual acuity, or other factors related to the risk of falling might be more attributable to the predisposition of fracture [33].

Limitations of this study are acknowledged. First, the WHI data we used only included women 50–79 years old, so our findings may not apply to men or to women who are not in the age range of this study. Second, genetic variants related to fracture risk independent of BMD remain mostly undiscovered and likely most related genetic variants have not been included in the present study. Therefore they had a limited impact on the T-score classification. Thirdly, concerning the allele frequencies, osteoporotic fracture risk is associated with common and rare variants. Since all SNPs used in the current study were based on a prior GWAS meta-analysis, which likely is able to discover only common genetic variants, the BMD or fracture-related rare genetic determinants may not be included. Finally, the sample size of minority subjects was very small in this study; the results may, therefore, be underpowered.

## 5. Conclusions

To the best of our knowledge, this is the first study to assess T-score performance in the prediction of MOF and any fractures in groups with different genetic profiling and of various races. Our findings demonstrated that T-score performed differently in different races and PGS groups, and thus the effect of race and genetic determinants in osteoporotic fracture prediction should be taken into account beyond the T-score classification. Fully integrating genetic profiling and racial factors into the existing fracture assessment model is very likely to improve the accuracy of osteoporosis diagnosis. Thus, developing racial/ethnic-specific, individualized osteoporosis diagnosis methods will provide more accurate fracture risk assessment and decrease false-positive rates and false-negative rates of osteoporosis diagnosis. Further studies, especially these including men, a more extensive sample of minorities, and more comprehensive fracture-associated genetic variants, are warranted.

## Figures and Tables

**Figure 1 jcm-09-00499-f001:**
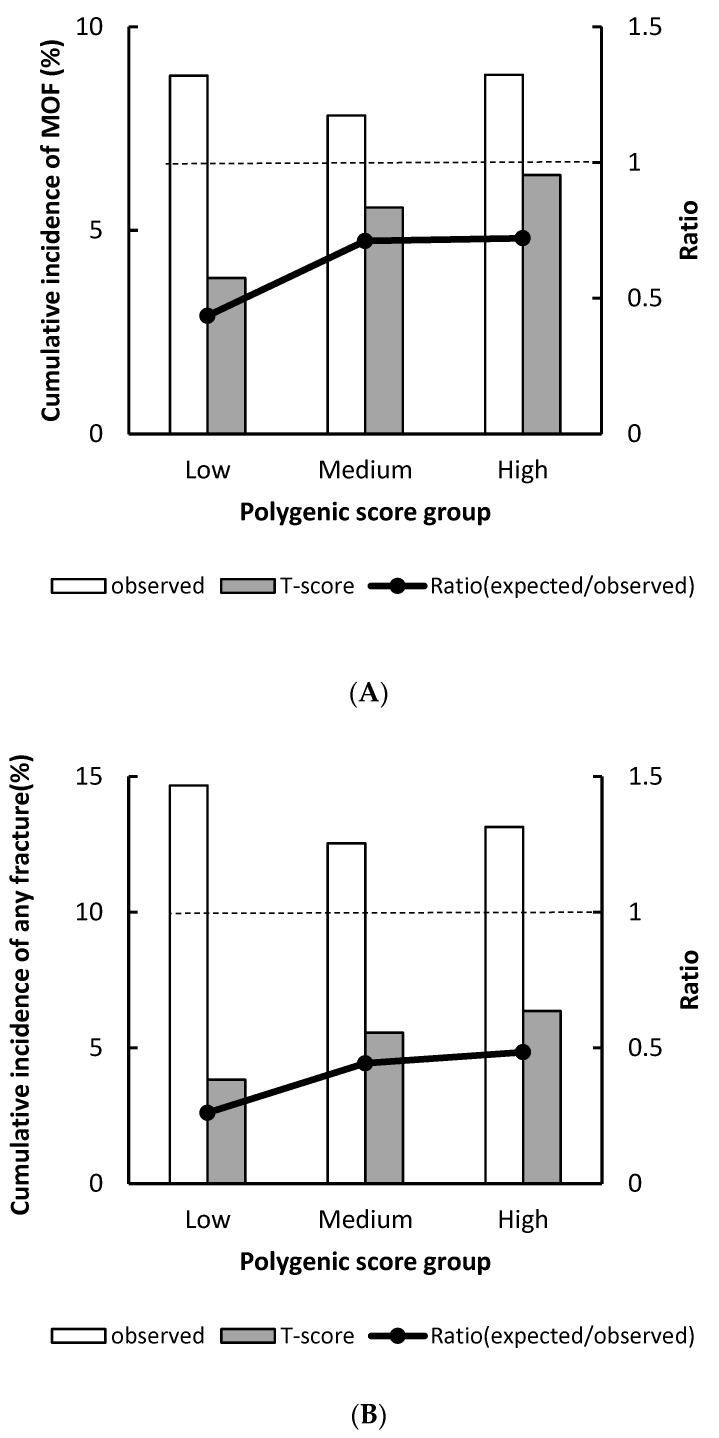
Observed versus predicted major osteoporotic fracture (**A**) and any fracture (**B**) probability stratified by polygenic score group. The dotted line indicates a relative ratio of 1 (reference line); ratio <1 indicates that T-score underestimates fracture risk.

**Figure 2 jcm-09-00499-f002:**
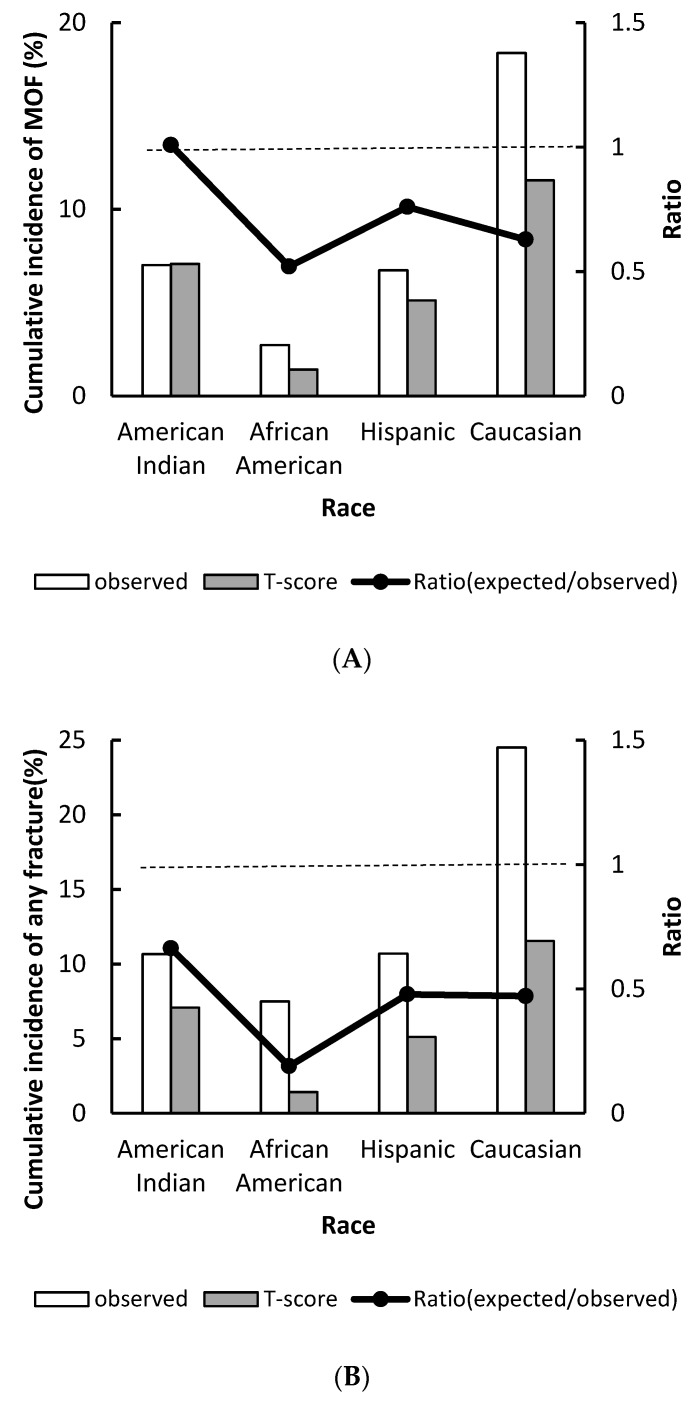
Observed versus predicted major osteoporotic fracture (**A**) and any fracture (**B**) probability stratified by race. The dotted line indicates a relative ratio of 1 (reference line), ratio <1 indicates that T-score underestimates fracture risk.

**Figure 3 jcm-09-00499-f003:**
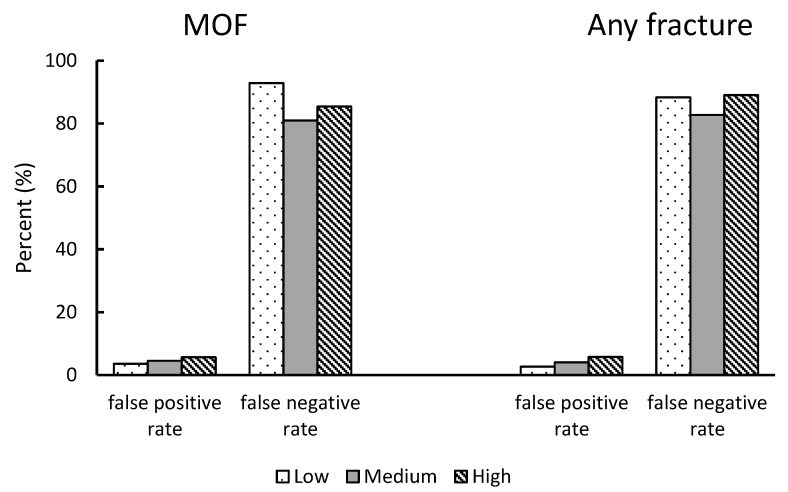
False-positive rate and false-negative rate for major osteoporotic fracture and any fracture in the analytic sample by polygenic score group.

**Figure 4 jcm-09-00499-f004:**
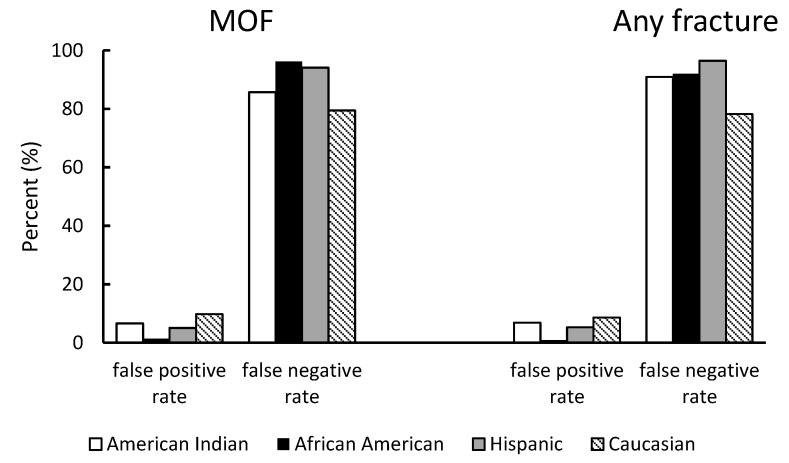
False-positive rate and false-negative rate for major osteoporotic fracture and any fracture in the analytic sample by race.

**Table 1 jcm-09-00499-t001:** Baseline characteristics of 2417 women with and without any subsequent fracture during 19 years of follow-up.

	Subjects with Any Fracture Event (n = 289)	Subjects without Any Fracture Event (n = 2128)	*p* Value
**Age (year), mean (SD)**	65.90 ± 7.33	62.38 ± 7.63	**<0.0001**
**Weight (kg), mean (SD)**	76.15 ± 15.59	77.56 ± 16.53	0.23
**Height (cm), mean (SD)**	161.61 ± 5.79	160.95 ± 6.01	0.12
**Body mass index (kg/m^2^), mean (SD)**	29.16 ± 5.58	29.89 ± 5.97	0.08
**Smoking, No. (%)**			**0.02**
**Never**	166 ± 59.29	1191 ± 56.74	
**Past**	100 ± 35.71	692 ± 32.397	
**Current**	14 ± 5.00	216 ± 10.29	
**≥3 Alcoholic Drinks per Day**			0.72
**Yes**	1 ± 0.35	18 ± 0.85	
**No**	288 ± 99.65	2110 ± 99.15	
**Rheumatoid Arthritis, No. (%)**			0.06
**Yes**	30 (10.38)	154 (7.24)	
**No**	259 (89.62)	1974 (92.76)	
**Previous Fragility Fractures, No. (%)**			**<0.0001**
**Yes**	114 (39.45)	528 (24.81)	
**No**	175 (60.55)	1600 (75.19)	
**Familial History of Hip Fracture, No. (%)**			**0.002**
**Yes**	40 (13.84)	177 (8.32)	
**No**	249 (86.16)	1951 (91.68)	
**PGS, mean (SD)**	2.27 ± 0.22	2.27 ± 0.23	0.81
**T-score, mean (SD)**	−1.36 ± 1.11	−0.72 ± 1.16	**<0.0001**
**Follow up days (SD)**	1967.19 ± 1457.08	5075.27 ± 2144.22	**<0.0001**

PGS: polygenic score calculated based on 63 BMD-related SNPs. Significant results are in boldface.

**Table 2 jcm-09-00499-t002:** Hazard Ratios (HR) with 95% Confidence Interval (CI) for Outcomes of Incidence Fracture According to Polygenic Score Group, adjusted for T-score diagnosis, and race: Results from the Cox Proportional Hazard Model.

	Major Osteoporotic Fracture	Any Fracture
	HR (95 % CI)	HR (95 % CI)
**Adjusted for T-score diagnosis**		
**low**	1 (reference)	1 (reference)
**medium**	0.86 (0.68–1.09)	0.81 (0.65–1.00)
**high**	0.98 (0.75–1.28)	0.89 (0.70–1.13)

Significant results are in boldface.

**Table 3 jcm-09-00499-t003:** Hazard Ratios (HR) with 95% Confidence Interval (CI) for Outcomes of Incidence Fracture According to Race Group, adjusted for T-score diagnosis, and Polygenic Score Groups: Results from Cox Proportional Hazard Model.

	Major Osteoporotic Fracture	Any Fracture
	OR (95 % CI)	OR (95 % CI)
**Adjusted for T-score diagnosis**		
**Caucasian**	1 (reference)	1 (reference)
**American Indian**	**0.59 (0.35–0.99)**	**0.53 (0.32–0.88)**
**African American**	**0.41 (0.33–0.52)**	**0.44 (0.36–0.54)**
**Hispanic**	**0.45 (0.35–0.58)**	**0.46 (0.36–0.58)**
**Adjusted for T-score diagnosis + PGS**		
**Caucasian**	1 (reference)	1 (reference)
**American Indian**	**0.56 (0.33–0.97)**	**0.52 (0.31–0.87)**
**African American**	**0.41 (0.33–0.52)**	**0.44 (0.35–0.54)**
**Hispanic**	**0.44 (0.34–0.58)**	**0.46 (0.36–0.59)**

PGS: polygenic score calculated based on 63 bone mineral density-related SNPs. Significant results are in boldface.

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
