# Peer review of "Evaluating the Performance of the WHO International Reference Standard for Osteoporosis Diagnosis in Postmenopausal Women of Varied Polygenic Score and Race"

_jcm, 2020, doi:10.3390/jcm9020499_

Round 1

Reviewer 1 Report

The authors of “Evaluating the performance of the WHO international reference standard for osteoporosis diagnosis in postmenopausal women of varied polygenic score and race” calculated the polygenic score (PGS) from 63 BMD-associated single nucleotide 15 polymorphisms (SNPs) for each participant (WHI study, n=2417) and observed an underestimation with t score of the risk of major osteoporotic fracture in African American women and in women with low PGS. Thus, the authors concluded that BMD T- score performance varies significantly by race in postmenopausal women.

In the Methods section, were fractured vertebrae excluded in the L2-L4 dxa scan? The authors used femoral neck BMD associated SNPs for PGS, thus it would be of interest to differentiate the predictive value of PGS for hip fracture from the other types of fracture. Please add information on therapy. Did patients, with fracture at baseline, have any treatment for osteoporosis? This data may influence their risk of fracture. How many patients at baseline, without fracture, had a T-score diagnosis of osteoporosis? How many of this group had a new fracture? How many patients had corticosteroid at baseline and during follow-up? Corticosteroid augments the risk of fracture, but BMD is not always severely diminished. Thus, it may be a confounding factor. How many patients were osteopenic or with a normal BMD at baseline, in these two groups, does PGS predict fracture differently?

Author Response

In the Methods section, were fractured vertebrae excluded in the L2-L4 dxa scan?

Authors’ response: Whether the subjects who have fractured vertebrae were excluded from the L2-L4 DXA scan was not specified in the original WHI documents. To eliminate the effect of previous fracture on fracture prediction in the present research, as mentioned in the Statistical Analysis section, we conducted a sensitivity analysis by excluding participants who had previous fractures; the corresponding results were similar to the results observed in the main analysis. 

The authors used femoral neck BMD associated SNPs for PGS, thus it would be of interest to differentiate the predictive value of PGS for hip fracture from the other types of fracture.

Authors’ response: Following the reviewers’ recommendation, we conducted a multinomial logistic regression to assess whether the predictive value of PGS for hip fracture would be different from other types of fracture. Results show that the predictive value of PGS for hip fracture is not different from the predictive value of PGS for non-hip fractures (p=0.51). (Please see the second paragraph under Statistic Analysis/Experimental Section and the first paragraph of PGS and the fracture outcome/Results section. A supplementary table 3 is also attached to the Appendix.)

Please add information on therapy. Did patients, with fracture at baseline, have any treatment for osteoporosis? This data may influence their risk of fracture.

Authors’ response: A restrict exclusion criteria was applied as participants who reported taking any medication known to influence osteoporosis, including bisphosphonates, calcitonin, parathyroid hormone, selective estrogen receptor modulators, luteinizing hormone-releasing hormone agents, and somatostatin agents were excluded from the study sample. The corresponding information has been added in the revision. (Please see the first paragraph of Participants/Experimental Section).

How many patients at baseline, without fracture, had a T-score diagnosis of osteoporosis? How many of this group had a new fracture?

Authors’ response: There were 52 participants free of previous fractures had a T-score diagnosis of osteoporosis at baseline, and 12 out of those 52 participants had a new fracture during the follow-up. The information is added to the manuscript accordingly. (Please see the first paragraph of Baseline characteristics/Results section)

How many patients had corticosteroid at baseline and during follow-up? Corticosteroid augments the risk of fracture, but BMD is not always severely diminished. Thus, it may be a confounding factor.

Authors’ response: The data of corticosteroid use was collected only in the first-year follow-up. In the current study sample, only 9 participants were taking corticosteroid; we, therefore, excluded them from the data analysis. This information is added to the Participants/Experimental Section accordingly.

How many patients were osteopenic or with a normal BMD at baseline, in these two groups, does PGS predict fracture differently? 

Authors’ response: Based on the T-score classification, there are 997 participants with osteopenia and 1291 participants with normal BMD at baseline. To determine whether PGS would predict fracture differently in osteopenia patients, or women with a normal BMD, we conducted subgroup analysis. Results from the analysis indicates that PGS does not predict fracture differently in these two groups (Table 1.). In the revision, this information is added to the second paragraph of the Statistical Analysis/Experimental Section and the sixth paragraph of the Results section accordingly.

Table 1. Odds ratios (OR) with 95% Confidence Interval (CI) for outcome of incidence fracture according to PGS groups: results from Univariate logistic regression.

T-score classification

PGS group

OR (95% CI)

Osteopenia

low

1(reference)

med

0.75 (0.43-1.30)

high

0.71 (0.43-1.16)

BMD normal

low

1(reference)

med

0.99 (0.47-2.12)

high

1.39 (0.60-3.19)

Reviewer 2 Report

Authors analyzed genomic data in 2,417 women, who were originally enrolled in Women’s Health Initiative (WHI) study aged 50-79 years old. They demonstrated that T-score classification significantly underestimated the risk of either major osteoporotic fracture or any other fractures, particularly in African American women and in women with low polygenic score.

1, subjects were enrolled at least 20 years ago. FRAX was not available at that time, but authors should estimate the fracture risk of each individuals analyzed in this study at the time of enrollment by FRAX, should examine the relationship/association between FRAX score (the ten year probability of fracture) and cumulative incidence of fractures, and T-score among races.

2, authors should demonstrate the PGS in each race.

3, authors should demonstrate the substituting reliable factor(s) other than or beyond T-score to estimate the risk of fractures among races.

Author Response

Subjects were enrolled at least 20 years ago. FRAX was not available at that time, but authors should estimate the fracture risk of each individuals analyzed in this study at the time of enrollment by FRAX, should examine the relationship/association between FRAX score (the ten-year probability of fracture) and cumulative incidence of fractures, and T-score among races.

Authors’ response: As mentioned in the Introduction section, we have already examined the performance of FRAX by race and pyogenic score in a previous peer-reviewed paper (Wu, Q., Xiao, X. & Xu, Y. (2020) Performance of FRAX in Predicting Fractures in US Postmenopausal Women with Varied Race and Genetic Profiles, Journal of Clinical Medicine. 9.). In that study, we successfully demonstrated that FRAX performs differently in different races, and the effect of race in osteoporotic fracture prediction has not heretofore adequately been taken into account by the existing FRAX model. Therefore, in this study, we focused on evaluating the performance of the T-score in osteoporotic fracture prediction in women of varied polygenic score and race.

Authors should demonstrate the PGS in each race.

Authors’ response: The mean PGS in American Indians, African Americans, Hispanics, and Caucasians were 2.42, 2.19, 2.41, and 2.25, respectively. Results from the ANOVA test shows that the difference between race groups were significant (p<.0001). This information is added to the first paragraph of the Statistical Analysis/Experimental Section and the first paragraph of the Results section accordingly.

Authors should demonstrate the substituting reliable factor(s) other than or beyond T-score to estimate the risk of fractures among races.

Authors’ response: A separate multivariate cox proportional hazard model were further employed to assess the effect of race on the outcome of MOF or any fracture within 19 years, with baseline T-score, age, BMI, and previous fracture controlled for. This information is added to the second paragraph of the Statistical Analysis/Experimental Section and the seventh paragraph of the Result sections accordingly. A supplementary table 4 is also attached to the Appendix.

Round 2

Reviewer 1 Report

The authors answered the questions

Reviewer 2 Report

Authors addressed the concerns raised by this reviewer.